# Effect of Additive Ti_3_SiC_2_ Content on the Mechanical Properties of B_4_C–TiB_2_ Composites Ceramics Sintered by Spark Plasma Sintering

**DOI:** 10.3390/ma13204616

**Published:** 2020-10-16

**Authors:** Xingheng Yan, Xingui Zhou, Honglei Wang

**Affiliations:** Science and Technology on Advanced Ceramic Fibers and Composites Laboratory, College of Aerospace Science and Engineering, National University of Defense Technology, Changsha 410073, China; yan-x-h@outlook.com (X.Y.); honglei.wang@163.com (H.W.)

**Keywords:** spark plasma sintering, boron carbide, Ti_3_SiC_2_, fracture toughness

## Abstract

B_4_C–TiB_2_ composite ceramics with ultra-high fracture toughness were successfully prepared via spark plasma sintering (SPS) at 1900 °C using B_4_C and Ti_3_SiC_2_ as raw materials. The results showed that compared with pure B_4_C ceramics sintered by SPS, the hardness of B_4_C–TiB_2_ composite ceramics was decreased, but the flexural strength and fracture toughness were significantly improved; the fracture toughness especially was greatly improved. When the content of Ti_3_SiC_2_ was 30 vol.%, the B_4_C–TiB_2_ composite ceramic had the best comprehensive mechanical properties: hardness, bending strength and fracture toughness were 27.28 GPa, 405.11 MPa and 18.94 MPa·m^1/2^, respectively. The fracture mode of the B_4_C–TiB_2_ composite ceramics was a mixture of transgranular fracture and intergranular fracture. Two main reasons for the ultra-high fracture toughness were the existence of lamellar graphite at the grain boundary, and the formation of a three-dimensional interpenetrating network covering the whole composite.

## 1. Introduction

Boron carbide is an attractive engineering material with a high melting point, low density, high hardness, high thermal conductivity and a large neutron absorption surface, which makes it a candidate material for wear-resistant parts, cutting tools, light armor products and neutron radiation shielding [1,2]. However, its low sintering property (due to the strong B–C covalent bond and B_2_O_3_ oxide layer) and poor fracture toughness limits its excellent performance. Spark plasma sintering (SPS) is a kind of electric current-assisted sintering technology, which can enhance the bonding and densification of particles through the combination of mechanical pressure, an electric field and a thermal field [3,4]. SPS adopts the same stamping/die system concept as hot pressing, although the heating methods are fundamentally different. Hot pressing sintering is heated by heater radiation, while the SPS heat source is Joule heat generated by the current of the mold or sample [5,6]. A heating rate of up to 1000 °C/min can be obtained by SPS, and the heating-up time is greatly shortened, which is beneficial to limit the grain growth [7]. In addition, currents can also enhance powder sintering by activating one or more parallel mechanisms, such as surface oxide removal, electromigration and electroplasticity [8].

Reasonable use of additives can stimulate boron carbide densification without any deterioration of mechanical properties. Some additives can react with boron carbide in situ to form nonvolatile second phases, which is helpful for densification and can enhance properties. Ti_3_SiC_2_ can react with B_4_C to form TiB_2_ with high hardness and a high melting point, which can be used as an ideal toughening phase for B_4_C ceramics [9]. In this study, B_4_C–TiB_2_ composite ceramics with ultra-high toughness were prepared by the SPS process with different contents of Ti_3_SiC_2_ as additives, and the influence mechanism of Ti_3_SiC_2_ content on the microstructure and properties of B_4_C–TiB_2_ composite ceramics was studied.

## 2. Experimental Procedure

Commercially available B_4_C powders (purity 99.9%, 1 μm, 4.53 g/cm^3^, Nangong Naiyate Alloy Welding Material Co., Ltd., Nangong, China) and Ti_3_SiC_2_ powders (purity 99.9%, <74 μm, 4.53 g/cm^3^, Nanjing Mingchang New Material Co., Ltd., Nanjing, China) were used as raw materials. Ti_3_SiC_2_–B_4_C powders containing 20 vol.%, 25 vol.%, 30 vol.% and 35 vol.% Ti_3_SiC_2_, respectively, were mixed for 24 h through a small vertical mixer at 80 r/min without adding solvent. Samples were prepared by SPS equipment (HP D 25/4-SD, FCT Systeme GmbH, Frankenblick, Germany) in a vacuum with 35 MPa mechanical pressure at 1900 °C for 5 min. The heating rate was 100 °C/min and the cooling rate was 50 °C/min. 

The absolute density of B_4_C–TiB_2_ composite ceramics was determined using the Archimedes method. Hardness was measured by a Vickers indentation tester (HMV-2TADW E, Shimadzu, Kyoto, Japan) at a 9.81 N load with a holding time of 15 s on the polished surface. Flexural strength was determined by a three-point bending test with a span of 30 mm and a loading speed of 0.5 mm/min, and the specimens used in the test were 3 mm × 4 mm × 35 mm bars. The SENB (Single-Edge Notched Beam) method was used to determine the fracture toughness of the specimens, with dimensions of 2 mm × 4 mm × 20 mm (with 2 mm high notch). The microstructures of the composite ceramics were characterized by X-Ray powder diffraction (XRD, X′ Pert PRO-MPD, Holland Panalytical, Almelo, Netherlands), scanning electron microscope (SEM, S-4800N, Hitachi, Tokyo, Japan), transmission electron microscope (TEM, JEM-2100, JEOL, Tokyo, Japan) and energy dispersive spectrometer (EDS, INCA, OXFORD INSTRUMENTS, Oxford, UK).

## 3. Results and Discussion

The scanning electron microscope (SEM) images and X-ray diffraction (XRD) patterns of the as-received powders of B_4_C and Ti_3_SiC_2_ are shown in Figure 1. The SEM images show that B_4_C particles have ladder-like surface undulation, a typical transgranular fracture which appears during the particle crushing process; Ti_3_SiC_2_ particles have an obvious lamellar structure. It can be seen from the XRD images that the two kinds of powders are relatively pure and almost no oxide exists (the content of oxide is too small to be detected in XRD). Figure 2 shows the phase composition of B_4_C–TiB_2_ ceramic composites prepared at 1900 °C with a different content of additive Ti_3_SiC_2_. There is no diffraction peak of Ti_3_SiC_2_ in any of the XRD images, which indicates that Ti_3_SiC_2_ had completely reacted with B_4_C. When the temperature is above 1200 °C, the following reactions occur [10]:(1)$$B_{4}C+{\mathrm{Ti}}_{3}{\mathrm{SiC}}_{2}\to 2{\mathrm{TiB}}_{2}+\mathrm{TiC}+\mathrm{SiC}+C$$
(2)$$B_{4}C+2\mathrm{TiC}\to 2{\mathrm{TiB}}_{2}+3C$$

The reactions (1) and (2) ended at 1600 °C, and based on the above results, the overall reaction in the system can be described as the following reaction [10]:(3)$$3B_{4}C+2{\mathrm{Ti}}_{3}{\mathrm{SiC}}_{2}\to 6{\mathrm{TiB}}_{2}+2\mathrm{SiC}+5C$$

TiC appears as an intermediate product in the whole reaction process but does not exist in the final product. According to the XRD test results, the content of each phase is shown in Table 1. TiB_2_ and B_4_C are the main phase composition of the composites, and a small amount of SiC and C exist. With the increase in Ti_3_SiC_2_ content, the proportion of TiB_2_, B_4_C and C (graphite) in the composite increases, while the content of B_4_C decreases. 

Table 2 shows the properties of the samples prepared with different contents of Ti_3_SiC_2_. In our experiment, the pure B_4_C samples sintered by SPS were broken when they were taken out of the mold, so the pure B_4_C sample BT0 prepared by Direct Current Sintering at 1800 °C for 10 min was used as the control group. BT0 was the reference sample without Ti_3_SiC_2_, and its hardness, bending strength and fracture toughness were 33.5 GPa, 224.43 MPa and 5.96 MPa·m^1/2^, respectively. Compared with BT0, all of the samples containing Ti_3_SiC_2_ had a higher relative density. Figure 3 shows the BSE (Backscattered Electron) images of BT0 and BT30 after polishing. There were some closed pores in BT0, but not in BT30. This is because the B_4_C particles have sharp edges and corners, and its hardness (55 GPa) is very high. Thus, they cannot be extruded and deformed under pressure, leaving a non-contact space inside, and form pores. The hardness of Ti_3_SiC_2_ (4 GPa) was much smaller than that of B_4_C; Ti_3_SiC_2_ can be extruded and deformed without leaving voids between particles under an external load. After reaction sintering, TiB_2_, B_4_C, SiC and C (existing in the form of graphite) in the composites have different thermal expansion coefficients, which makes the ceramics more compact after cooling. 

The hardness of the second phase particles produced by the reaction is lower than that of B_4_C, especially the graphite phase, therefore it is inevitable that the hardness of the composite ceramics is lower than that of the pure B_4_C ceramics. Compared with BT0, the flexural strength and fracture toughness of BT20–BT35 are improved, and the hardness decreases. Figure 4a,b show the fracture morphology of BT0 and BT30, respectively. It can be seen that the fracture surface of BT0 is flat, which is a typical transgranular fracture morphology. Comparatively, the fracture surface of sample BT30 is rough, which is a typically mixed fracture morphology of transgranular fracture and intergranular fracture. In Figure 4b, the dark gray flat area is the B_4_C matrix, and the light gray rough area is TiB_2_ particles, which indicates that the fracture modes of the B_4_C phase and the TiB_2_ phase are transgranular fracture and intergranular fracture, respectively. In addition, there is a dark gray lamellar phase around the TiB_2_ grain, which can be inferred ad carbon phase by energy spectrum analysis. Figure 4c–e show the energy spectrum of B_4_C, TiB_2_ and C, respectively. Due to the mismatch of thermal expansion coefficients between B_4_C, TiB_2_ and graphite (B_4_C: 4.5 × 10^–6^ k^–1^; TiB_2_: 8.1 × 10^–6^ k^–1^; Graphite: 1 × 10^–6^ k^–1^ in the parallel direction, 29 × 10^–6^ k^–1^ in c direction) [11], there will be large residual stress at the interface of the phases, which will induce crack deflection along the grain boundary and extend the crack propagation path to improve the strength and toughness of the material. The nano TiB_2_ particles embedded in the B_4_C matrix will introduce internal stress, which will strengthen the B_4_C matrix by a lattice distortion effect, and can also nail the dislocations and hinder their movement, so as to enhance the strength of the material.

It should be noted that the fracture toughness of the composites was greatly improved by adding more than 20 vol.% of Ti_3_SiC_2_. The fracture toughness of BT30 was 18.94 MPa·m^1/2^, which was more than three times of that of BT0 (5.80 MPa·m^1/2^), and more than twice as high as the highest fracture toughness cited in Table 3. Wen Q et al. [9] adopted the same ratio of raw materials as ours to sinter the ceramics. They used 0.5 μm B_4_C powders and 0.5–10 μm Ti_3_SiC_2_ powders as raw materials, and their process was 1850 °C hot pressing sintering for 30 min. Compared with their work, the particle size of the raw powders we used had a greater difference in size (B_4_C: 1 μm; Ti_3_SiC_2_: <74 μm), meaning aggregates were formed more easily, which are beneficial for toughness but unfavorable for strength [12]. Besides, our ceramics had a higher relative density and graphite content, which may be due to the evaporation of Si at a higher sintering temperature, and the electric field [13]. In the analysis of XRD data, only graphite matched with the existing peaks, and the diffraction peaks of other existing forms of carbon did not correspond with the peaks in the experimental data. Moreover, the sintering temperature was 1900 °C, at which point the amorphous carbon would also be graphitized. Therefore, we infer that the lamellar phase in Figure 4b is graphite. Graphite phase exists at the grain boundaries of TiB_2_ and B_4_C, which reduces the bonding strength of the interface and has an adverse effect on the hardness and strength of the composite ceramics. This is the reason why the flexural strength of BT30 is at a low level in Table 3. At the same time, the existence of graphite can limit grain growth. In the cooling process, microcracks are produced under the effect of interfacial stress produced by different thermal expansion coefficients, and the cracks propagate along the interlayer of graphite during fracture, resulting in lamellar pull-out. This lamellar pull-out process greatly prolongs the path of crack propagation, consumes the energy of cracks, and is beneficial to improving the fracture toughness. It can be seen from Figure 4b that there are traces of particle pull-out and lamellar graphite pull-out in the fracture surface of BT30, which is one of the important reasons for the toughening of the composite. 

Figure 5 shows the variation of the relative density and hardness of B_4_C–TiB_2_ composite ceramics with the content of Ti_3_SiC_2_. Due to the slight evaporation of silicon [13], the density of the composites is slightly higher than the theoretical density. The evaporation capacity of Si increases with the increase in Ti_3_SiC_2_ content, resulting in an increase in the relative density. The hardness decreases with the increase in Ti_3_SiC_2_ content, because the proportion of the second phase with lower hardness, especially graphite, increases. 

Figure 6 shows the variation of flexural strength and fracture toughness of B_4_C–TiB_2_ composite ceramics with additive Ti_3_SiC_2_ content. When the content of Ti_3_SiC_2_ is in the range of 20 vol.%–30 vol.%, the flexural strength and fracture toughness are positively correlated with the content of Ti_3_SiC_2_. Comparing Figure 7a–c, it can be seen that the region as shown in Figure 7e becomes larger and more numerous with the increase in Ti_3_SiC_2_ content. In this region, TiB_2_, graphite and B_4_C intersect each other to form a three-dimensional interpenetrating network. During the crack propagation process, multiple two-phase interfaces must be bypassed to disperse into more small cracks and a large number of changes in the propagation direction. The pull-out mechanism of graphite also plays an important role in this agglomeration area. However, this area does not exist in isolation. Every small network links to each other, forming a large network structure covering the whole composite. At the same time, the network divides the B_4_C concentration area into small parts and surrounds them, so that there is no large area of a continuous B_4_C phase in the composites. The large area of a continuous B_4_C phase is very unfavorable to the toughness of the composite. With the increase in interlacing degrees of TiB_2_, SiC, graphite and B_4_C, the cracks need to bypass more two-phase interfaces, change direction more often, and disperse into more small cracks, which can consume a lot of energy of crack propagation and hinder the spreading of cracks. Therefore, the fracture toughness of B_4_C–TiB_2_ composite ceramics is greatly improved by the overall three-dimensional interpenetrating network structure.

However, when the content of Ti_3_SiC_2_ is 35 vol.%, as shown in Figure 7d, a large TiB_2_–SiC agglomerated area appears in the BT35 (among which the dark gray, medium gray and light gray phases are B_4_C, SiC and TiB_2_, respectively). There is a bad stress effect in the multiphase mixing region with more SiC. As shown in Figure 7f, many microcracks due to mismatching of the thermal expansion coefficient of SiC and TiB_2_ can be found in this region. These microcracks are conducive to the toughening of the material, but result in a significant decrease in the bending strength.

## 4. Conclusions

Ultra-high toughness B_4_C–TiB_2_ composite ceramics were prepared by the SPS method at 1900 °C. The content of additive Ti_3_SiC_2_ has a great influence on the microstructure and mechanical properties. With the increase in Ti_3_SiC_2_ content, the relative density and fracture toughness of the material increase, while the hardness decreases, and the flexural strength first increases and then decreases. When the content of Ti_3_SiC_2_ is 30 vol.%, B_4_C–TiB_2_ composite ceramics have the highest bending strength and the best comprehensive mechanical properties: hardness 27.28 GPa, bending strength 405.11 MPa, and fracture toughness 18.94 MPa·m^1/2^. The fracture mode of the material is a mixture of transgranular fracture and intergranular fracture. The existence of the graphite phase has a negative effect on the hardness and flexural strength of B_4_C–TiB_2_ composite ceramics, but it is beneficial to the fracture toughness. The main reason for obtaining high fracture toughness is the formation of a three-dimensional interpenetrating network covering the whole composite. 

## Figures and Tables

**Figure 1 materials-13-04616-f001:**
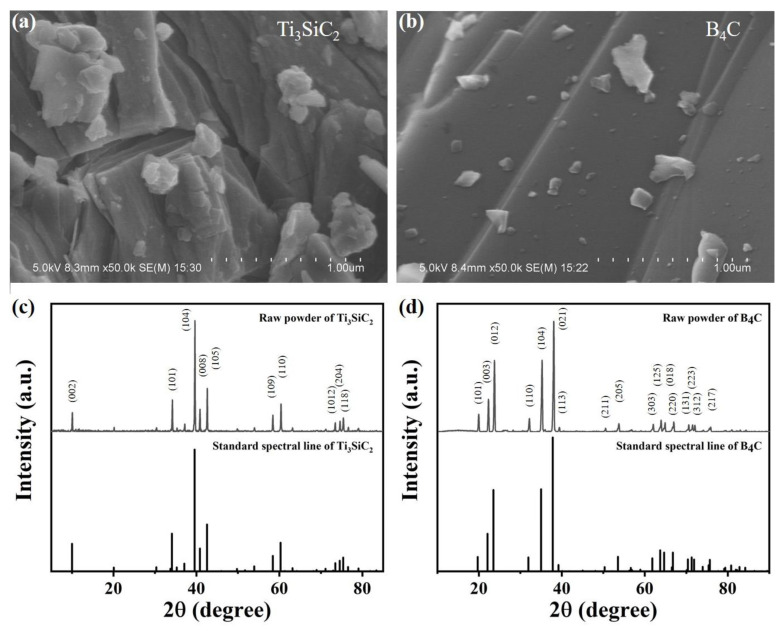
(**a**) SEM image of Ti_3_SiC_2_; (**b**) SEM image of B_4_C; (**c**) XRD pattern Ti_3_SiC_2_; (**d**) XRD pattern of B_4_C.

**Figure 2 materials-13-04616-f002:**
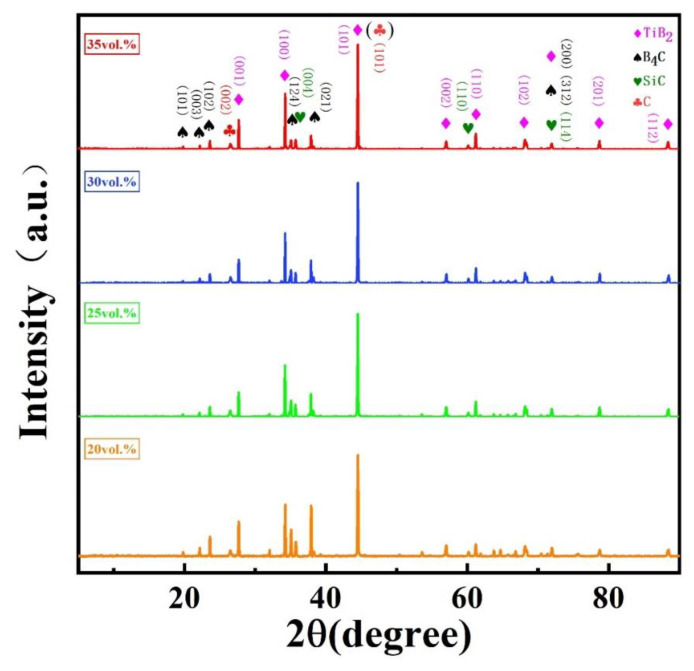
XRD patterns of B_4_C–TiB_2_ ceramic composites sintered at 1900 °C with different contents of the additive Ti_3_SiC_2_.

**Figure 3 materials-13-04616-f003:**
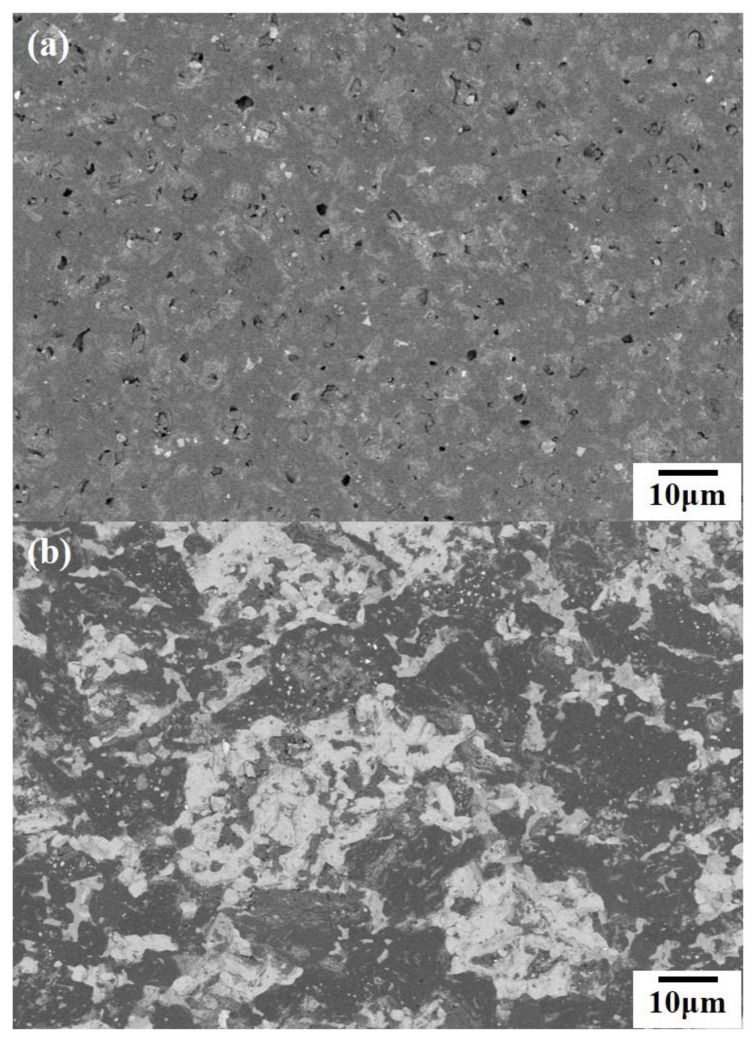
(**a**) BSE image of BT0; (**b**) BSE image of BT30.

**Figure 4 materials-13-04616-f004:**
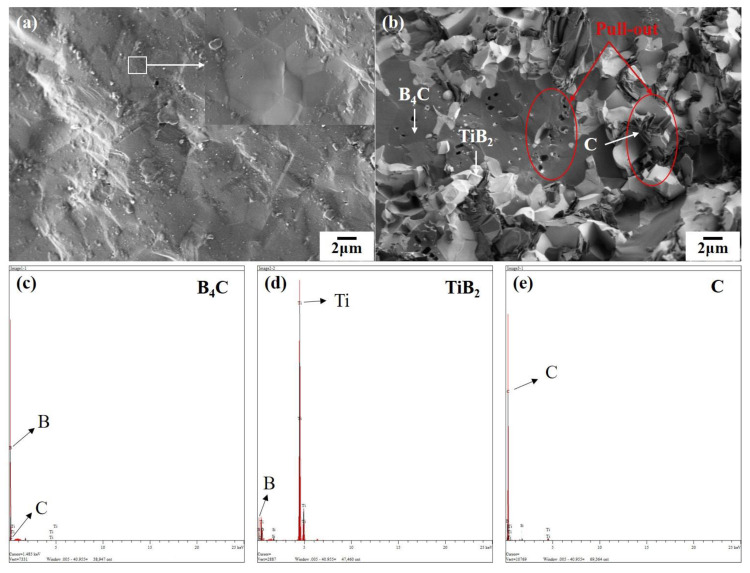
(**a**) SEM image of fracture surface of BT0; (**b**) SEM image of fracture surface of BT30; (**c**) Energy spectrum of B_4_C; (**d**) Energy spectrum of TiB_2_; (**e**) Energy spectrum of C.

**Figure 5 materials-13-04616-f005:**
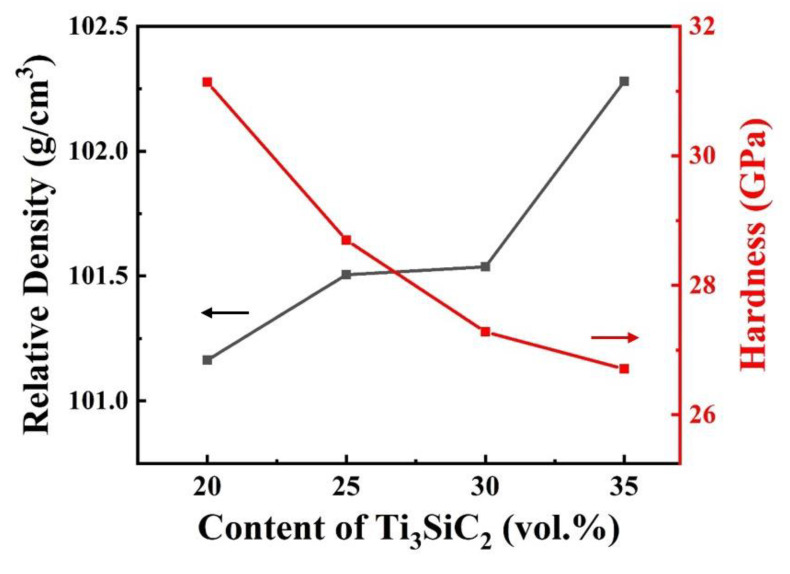
Relative density and hardness of the B_4_C–TiB_2_ composite ceramics sintered with different contents of Ti_3_SiC_2_.

**Figure 6 materials-13-04616-f006:**
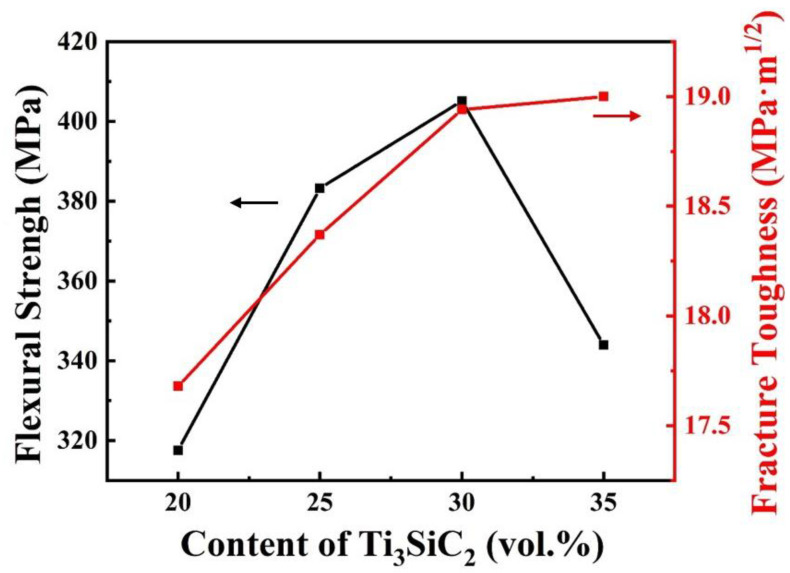
Flexural strength and fracture toughness of the B_4_C–TiB_2_ composite ceramics sintered with different contents of Ti_3_SiC_2_.

**Figure 7 materials-13-04616-f007:**
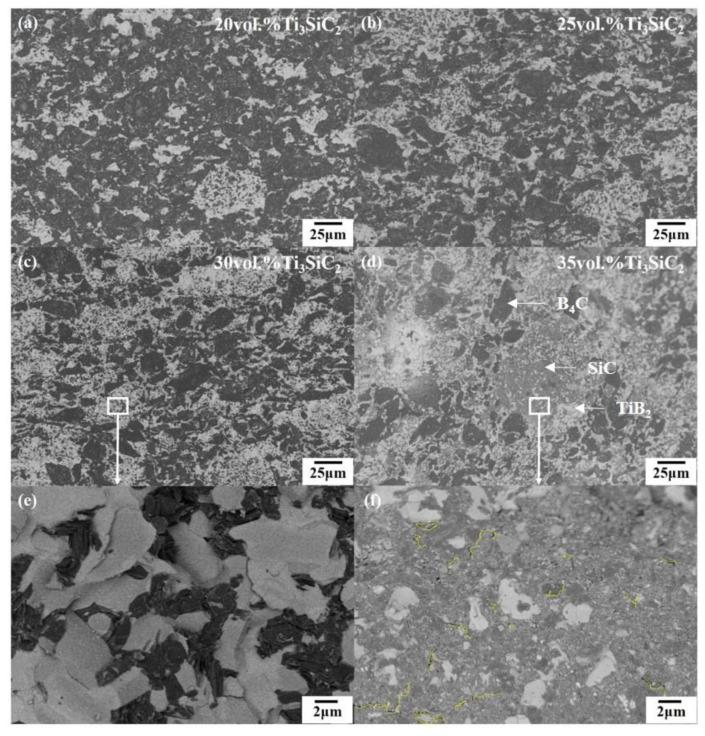
(**a**–**d**) BSE images of B_4_C–TiB_2_ composite ceramics sintered with 20 vol.%, 25 vol.%, 30 vol.%, 35 vol.% Ti_3_SiC_2_ in order; (**e**) Local magnification BSE image of (**c**); (**f**) Local magnification BSE image of (**d**).

**Table 1 materials-13-04616-t001:** Contents of different phases in B_4_C–TiB_2_ composite ceramics sintered at different temperatures.

Sample Name	Content of Ti_3_SiC_2_ (vol.%)	TiB_2_ (wt.%)	B_4_C (wt.%)	SiC (wt.%)	C (Graphite) (wt.%)
BT20	20	9.4	87.4	1.6	1.7
BT25	25	19.7	73.2	3.3	3.8
BT30	30	29.6	61.9	4.2	4.4
BT35	35	40.6	47.7	6.1	5.6

**Table 2 materials-13-04616-t002:** Properties of ceramics prepared with different contents of additive Ti_3_SiC_2_.

Simple Name	Content of Ti_3_SiC_2_ (vol.%)	Density (g/cm^3^)	Relative Density (%)	Hardness (GPa)	Flexural Strengh (MPa)	Fracture Toughness (MPa·m^1/2^)
BT0	0	2.50	99.20	33.50	224.43	5.96
BT20	20	3.12	101.16	31.14	317.55	17.68
BT25	25	3.13	101.51	28.70	383.25	18.37
BT30	30	3.17	101.54	27.28	405.11	18.94
BT35	35	3.17	102.28	26.71	343.95	19.00

**Table 3 materials-13-04616-t003:** Comparison of the properties of the B_4_C–TiB_2_ composite ceramics reported in recent years.

Serial No.	Starting Powder	Relative Density (%)	K_IC_, (MPa·m^1/2^)	Flexural Strength (MPa)	Ref. (Year)
1	B_4_C + 5 wt% (Ti_3_SiC_2_ + Si)	-	5.61	457.6	[14] (2019)
2	B_4_C + 30 wt% (TiB_2_ + Si)	99.6	5.77	531.2	[15] (2018)
3	B_4_C + 20 mol%TiB_2_	97.9	3.7	-	[16] (2020)
4	B_4_C + 15 wt%SiC + 20 mol%TiB_2_	98.6	4.2	343.8	[17] (2020)
5	B_4_C + 6.45 vol.%SiC + 7.78 vol.%TiB_2_	99.62	6.38	632	[12] (2019)
6	B_4_C + 30 vol.% Ti_3_SiC_2_	98.72	8.0	492.3	[9] (2017)
BT30	B_4_C + 30 vol.% Ti_3_SiC_2_	101.54	18.94	405.11	This work

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
