# Peer review of "Effect of Additive Ti3SiC2 Content on the Mechanical Properties of B4C–TiB2 Composites Ceramics Sintered by Spark Plasma Sintering"

_materials, 2020, doi:10.3390/ma13204616_

Round 1
Reviewer 1 Report
Effect of additive Ti3SiC2 content on mechanical properties of B4C–TiB2 composites ceramics sintered by spark plasma sintering
In this paper the authors have presented their work on the mechanical characterization of B4C-TiB2 composite ceramics synthesized using raw powders of B4C and Ti3SiC2 via spark plasma sintering method. Author observed that Ti3SiC2 percentage has very strong effect on the microstructure and mechanical properties of composite ceramics. By increasing the Ti3SiC2 fraction into B4C the hardness decreased, but the flexural strength and fracture toughness significantly improved, with 30 vol. % Ti3SiC2 displayed the best mechanical properties. Authors explained this change of property due to formation of a three dimensional interpenetrating network of phases of B4C, TiB2, SiC, graphite, and the existence of lamellar graphite at the grain boundary providing added strength and toughness. The authors further described the fracture mode of the composite as a mixture of transgranular fracture and intergranular fracture.
Comments:
- Please label crystallographic planes in XRD patterns of Figure 1 and Figure 2 (major one).
- Please enlarge important Figure 2 with major crystallographic planes labeled.
- Line 78-79: "TiC appears as an intermediate product in the whole reaction process but does not exist in the final product." What is the evidence for this? What do you mean by appears? Is the any reference paper?
- Table 2: How the reference sample BT0 was synthesized? SPS?
- Line 109-112: Please show/indicate this description by arrow on Figure 4 (a) and (b)
- Line 113: "In addition, there is a dark gray lamellar phase around the TiB2 grain, which can be inferred as graphite phase by energy spectrum analysis."
Where is the energy spectrum?
- Line 136-138: "It can be seen from Fig.4 (b) that there are traces of particle pull-out and lamellar graphite pull-out in the fracture surface of BT30, which is one of the important reasons for the toughening of the composite."
Please show/indicate this description by arrow on Figure 4 (b)
- Line 163: the toughness of the composite. With the increase of interlacing degree of TiB2, graphite and B4C, the...
Do you mean hardness and not toughness?
Do you mean SiC and not B4C?
- Line 172-173: "There is a bad stress effect in the multiphase mixing region with more SiC."
What is the meaning of this statement?
- Line 187-189: "The main reason for obtaining high fracture toughness is the existence of the graphite phase and the formation of a three-dimensional interpenetrating network covering the whole composite."
Authors made conclusions about effect of graphite on the mechanical behavior of composites without giving any clear evidence (spectroscopy) of presence of graphite phase!

Author Response
Dear Reviewer,
Your comments were highly insightful and enabled us to greatly improve the quality of our manuscript. In the following pages are our point-by-point responses to each of the comments of you. And in the attached manuscript, all the problems mentioned by the reviewers have been modified, so some changes have been made in the places you didn't mention.
Comments:
- Please label crystallographic planes in XRD patterns of Figure 1 and Figure 2 (major one).
Response: Figure 1 has been modified.
- Please enlarge important Figure 2 with major crystallographic planes labeled.
Response: Figure 2 has been modified.
- Line 78-79: "TiC appears as an intermediate product in the whole reaction process but does not exist in the final product." What is the evidence for this? What do you mean by appears? Is the any reference paper?
Response: This conclusion is quoted from the references, and the specific reaction process has been supplemented in the paper.
- Table 2: How the reference sample BT0 was synthesized? SPS?
Response: BT0 was prepared by Direct Current Sintering at 1800 ℃ for 10 min, which has been supplemented in this paper.
- Line 109-112: Please show/indicate this description by arrow on Figure 4 (a) and (b)
Response: Figure 4 has been modified.
- Line 113: "In addition, there is a dark gray lamellar phase around the TiB2grain, which can be inferred as graphite phase by energy spectrum analysis."
Where is the energy spectrum?
Response: The energy spectrum data has been added in Figure 4.
- Line 136-138: "It can be seen from Fig.4 (b) that there are traces of particle pull-out and lamellar graphite pull-out in the fracture surface of BT30, which is one of the important reasons for the toughening of the composite."
Please show/indicate this description by arrow on Figure 4 (b)
Response: Figure 4 has been modified.
- Line 163: the toughness of the composite. With the increase of interlacing degree of TiB2, graphite and B4C, the...
Do you mean hardness and not toughness?
Do you mean SiC and not B4C?
Response: For the first question, there was ambiguity in the original manuscript. What we wanted to express was that the large area of continuous B4C phase is very unfavorable to the toughness of the composite. And this problem has been revised in the article.
For the second question, What we wanted to express was the interlacing degrees of different phases, including TiB2, SiC, graphite and B4C. This problem has been revised in the article.
- Line 172-173: "There is a bad stress effect in the multiphase mixing region with more SiC."
What is the meaning of this statement?
Response: The statement means the mismatch of the thermal expansion coefficient of TiB2 and SiC, which created many microcracks in the TiB2-SiC agglomerated area. This statement has been explained in combination with Figure 7(f) in the following.
- Line 187-189: "The main reason for obtaining high fracture toughness is the existence of the graphite phase and the formation of a three-dimensional interpenetrating network covering the whole composite."
Authors made conclusions about effect of graphite on the mechanical behavior of composites without giving any clear evidence (spectroscopy) of presence of graphite phase!
Response: We found the layered pullout structure by scanning electron microscope in the beginning, and then determined it is the carbon phase by the energy spectrum. In the analysis of XRD data, only graphite can find the existence of matching peaks, and the diffraction peaks of other existing forms of carbon can not correspond to the peaks in the experimental data. And the sintering temperature is 1900 ℃, at which the amorphous carbon will also be graphitized. These are the reasons why we think it is graphite.
It is true that spectral data will make it more convincing, but because of the limitations of experimental conditions and funds, the test was not carried out. And the first author needs this article to apply for a degree in the next few days, so we hope that we can complete the revision of this article without supplementary data.
We tried our best to improve the manuscript and made some changes These changes will not influence the content and framework of the paper. We appreciate your warm work earnestly and hope that the correction will meet with approval. Once again, thank you very much for your comments and suggestions.
Reviewer 2 Report
- Page 2, line 52. The samples should be instead of “simples”.
- Figure 2, page 3, line 74. The authors manifest that …” There is no diffraction peak of Ti3SiC2 in all XRD images”. The characteristic peak of Ti3SiC2 is a low angle 002 peak located at around 10 degree 2 theta (Figure 1c). However, the authors provide XRD data starting from 10 degree 2 Theta (Figure 2). That is why this is difficult to judge if the MAX phase is in the composite or not. To clarify this point, I suggest to start the XRD pattern from the angges presented in the Figure 1c.
- Page 8, line 171. “However, when the content of Ti3SiC2 is 35 vol.%, as shown in Fig. 6 (d), a large TiB2-SiC agglomerated area appears in the BT35 (among which the dark gray, medium gray and light gray phases are B4C, SiC and TiB2 respectively).”The authors indicated a wrong number of Figure. It should be Figure 7d instead of 6d. In addition, this is better to mark B4C, SiC and TiB2 phase on the images with arrows instead of describing their colors in the brackets.
Author Response
Dear Reviewer,
Your comments were highly insightful and enabled us to greatly improve the quality of our manuscript. In the following pages are our point-by-point responses to each of the comments of you. And in the attached manuscript, all the problems mentioned by the reviewers have been modified, so some changes have been made in the places you didn't mention.
Comments:
1. Page 2, line 52. The samples should be instead of “simples”.
Response:This issue has been revised.
2. Figure 2, page 3, line 74. The authors manifest that …” There is no diffraction peak of Ti3SiC2 in all XRD images”. The characteristic peak of Ti3SiC2 is a low angle 002 peak located at around 10 degree 2 theta (Figure 1c). However, the authors provide XRD data starting from 10 degree 2 Theta (Figure 2). That is why this is difficult to judge if the MAX phase is in the composite or not. To clarify this point, I suggest to start the XRD pattern from the angges presented in the Figure 1c.
Response: This problem has been revised in accordance with the reviewers' comments.
3. Page 8, line 171. “However, when the content of Ti3SiC2 is 35 vol.%, as shown in Fig. 6 (d), a large TiB2-SiC agglomerated area appears in the BT35 (among which the dark gray, medium gray and light gray phases are B4C, SiC and TiB2 respectively).”The authors indicated a wrong number of Figure. It should be Figure 7d instead of 6d. In addition, this is better to mark B4C, SiC and TiB2 phase on the images with arrows instead of describing their colors in the brackets.
Response: This problem has been revised in accordance with the reviewers' comments.
We tried our best to improve the manuscript and made some changes. These changes will not influence the content and framework of the paper. We appreciate your warm work earnestly and hope that the correction will meet with approval.
Once again, thank you very much for your comments and suggestions.
Round 2
Reviewer 1 Report
- Line 113: "In addition, there is a dark gray lamellar phase around the TiB2grain, which can be inferred as graphite phase by energy spectrum analysis."
Where is the energy spectrum?
Response: The energy spectrum data has been added in Figure 4.
New Comment 6: There is no description about energy spectrum Fig 4 (c), (d), and (e) in the paper. There is no figure caption below Figure 4. Authors must make these changes to the paper.
- Line 187-189: "The main reason for obtaining high fracture toughness is the existence of the graphite phase and the formation of a three-dimensional interpenetrating network covering the whole composite."
Authors made conclusions about effect of graphite on the mechanical behavior of composites without giving any clear evidence (spectroscopy) of presence of graphite phase!
Response: We found the layered pullout structure by scanning electron microscope in the beginning, and then determined it is the carbon phase by the energy spectrum. In the analysis of XRD data, only graphite can find the existence of matching peaks, and the diffraction peaks of other existing forms of carbon can not correspond to the peaks in the experimental data. And the sintering temperature is 1900 ℃, at which the amorphous carbon will also be graphitized. These are the reasons why we think it is graphite.
It is true that spectral data will make it more convincing, but because of the limitations of experimental conditions and funds, the test was not carried out. And the first author needs this article to apply for a degree in the next few days, so we hope that we can complete the revision of this article without supplementary data.
We tried our best to improve the manuscript and made some changes These changes will not influence the content and framework of the paper. We appreciate your warm work earnestly and hope that the correction will meet with approval. Once again, thank you very much for your comments and suggestions.
New Comment 10: If authors think that sufficient proof is not available in current situation, authors should change the conclusion focusing only on the formation of a three-dimensional interpenetrating network covering the whole composite. Authors might also suggest possibility of graphite phase contributing to the toughness in the discussion section.

Author Response
Dear Reviewer,
Your comments were highly insightful and enabled us to greatly improve the quality of our manuscript. In the following pages are our point-by-point responses to each of the comments of you. And in the attached manuscript, all the problems mentioned by you have been modified.
Comments:
NEW Comment 6: There is no description about energy spectrum Fig 4 (c), (d), and (e) in the paper. There is no figure caption below Figure 4. Authors must make these changes to the paper.
RESPONSE:we have added the description of energy spectrum Fig 4 (c), (d), and (e) in line 121, and the figure caption of Figure 4 has been modified. At the same time, labels have been added to the energy spectrum.
New Comment 10: If authors think that sufficient proof is not available in current situation, authors should change the conclusion focusing only on the formation of a three-dimensional interpenetrating network covering the whole composite. Authors might also suggest possibility of graphite phase contributing to the toughness in the discussion section.
RESPONSE:We have revised the conclusions in accordance with your comment, added the inference of the graphite in line 138-142,and supplemented the effect of graphite on toughness in line 148-150. The discussion on the effect of graphite on toughness in line 144-152 has been highlighted in red underline.
Once again, thank you very much for your comments and suggestions.
Reviewer 2 Report
The presentation of the paper is improved after author's correction
Author Response
Dear Reviewer,
Your comments were highly insightful and enabled us to greatly improve the quality of our manuscript.
We hope you would like to sign your review report.
Once again, thank you very much for your comments and suggestions.